# Recent Advances of Pd/C-Catalyzed Reactions

Zhenjun Mao *, Haorui Gu and Xufeng Lin *

Department of Chemistry, Zhejiang University, Hangzhou 310027, China; 0019529@zju.edu.cn
* Correspondence: maozhenjun@zju.edu.cn (Z.M.); lxfok@zju.edu.cn (X.L.); Tel.: +86-0571-8795-2759 (X.L.)

**Abstract:** The Pd/C-catalyzed reactions, including reduction reactions and cross-coupling reactions, play an irreplaceable role in modern organic synthesis. Compared to the homogeneous palladium catalyst system, the heterogeneous Pd/C catalyst system offers an alternative protocol that has particular advantages and applications. Herein, a review on Pd/C-catalyzed reactions is presented. Both the advances in Pd/C-catalyzed methodologies and the application of Pd/C-catalysis in total synthesis are covered in this review.

**Keywords:** palladium on carbon; heterogeneous catalysis; reduction; cross-coupling

## 1. Introduction

Palladium-catalyzed organic reactions are a fundamental tool of modern organic synthesis. Among all the known palladium-based catalysts, palladium on carbon (Pd/C) has attracted much attention due to its excellent catalytic activity in a variety of different organic reactions [1]. The application of Pd/C catalysis dates back to 1972, when Heck and co-workers first used the Pd/C catalyst system for the Heck coupling reaction [2]. Since then, Pd/C has proven to be very beneficial in cross-coupling reactions.

Palladium tightly bound to nanoparticle activated carbon has a large surface area and, thus, high catalytic efficiency. Repeated filtration processes are possible due to their high stability, so the Pd/C can be easily recycled and reused with little loss of activity. Their high tolerance to strong acids or bases, air, and moisture makes Pd/C catalysts suitable for a variety of reactions under moderate or harsh conditions [3,4]. Due to these advantages, Pd/C catalysts have not only attracted much attention in recent laboratory research, but are also becoming increasingly popular in industrial production.

Numerous works on ligand-free Pd/C-catalyzed reactions have been reported [5–10]. Ligand-free catalytic systems avoid the use of hazardous reagents, making the protocols much cleaner and environmentally friendly. However, in certain cases, ligands still play an important role; for example, in the presence of a ligand, the reaction rate and activity are much higher than in ligandless catalytic systems [11–16].

Although several reviews on Pd/C-catalyzed reactions have already been published in the last fifteen years [3,4,8,10], in this review, we examined recent advances in the chemistry of Pd/C-catalyzed reactions, such as reduction reactions and cross-coupling reactions, because these reactions are widely used for organic synthesis, and they also provide a guiding principle for designing the route to be used by organic chemists.

## 2. Catalyst

The abbreviation "Pd/C" stands for palladium metal on carbon support, an effective heterogeneous palladium catalyst. The carbon supported catalyst system has interesting features with respect to its application. Recovery, refining and recycling processes can be simplified with these heterogeneous Pd/C catalysts. Many properties of Pd/C, such as palladium's distribution, the oxidation state of palladium, its water content, etc., significantly affect the efficiency of the Pd/C catalyst.

Generally, three types of Pd/C, namely "egg-shell", "thick-shell", and "uniform", are found on charcoal with respect to palladium distribution [3]. Different depths of impregnation of palladium are found in the three catalyst types [17,18]. In the egg-shell catalyst, the palladium is densely distributed on the surface within 20–150 nm. In the uniform catalyst, the palladium is distributed uniformly. In the thick-shell catalyst, the palladium is distributed 200–500 nm away from the surface.

Pd/C catalysts are in the form of nanoparticles, since the catalytically active crystallite size should be between 2 nm and 20 nm [14,17]. Generally, commercially available Pd/C consists of both Pd(II) and Pd(0), the ratio depending on the method of preparation. Pure Pd(0) or Pd(II) is difficult to obtain, because Pd(0) is prepared through the reduction of Pd(II) and, inevitably, traces of Pd(II) remain in the catalytic system, while pure Pd(II) necessarily undergoes an intrinsic reduction process to produce Pd(0).

Charcoal can absorb a large amount of water, up to 55–60% by weight. For safety reasons, Pd/C containing a certain amount of water is necessary and useful, especially for large-scale industrial applications. For many Pd/C-catalyzed reactions, only commercially available Pd/C catalysts are needed, while, for certain reactions, a special production approach for the preparation of Pd/C is required. Methods for the preparation of Pd/C vary, resulting in different catalytic activities for different Pd/C catalysts in certain reactions. This also explains the poor reproducibility when some of the reported Pd/C-catalyzed reactions were carried out repeatedly [10].

Ligandless Pd/C-catalyzed reactions have been of great interest in recent years. A ligand-free synthetic system is considered clean, environmentally friendly and convenient and has been used for a variety of reactions, including reductions and cross-coupling reactions, while Pd/C with certain ligands, such as $PPh_3$, 1,1′-bis(diphenylphosphino)ferrocene (dppf), is still more effective in certain cases compared to ligand-free catalytic systems [8].

The standardized Pd/C catalyst should be denoted as X% Pd/C (Y mol%), where X is X wt% palladium on carbon and Y is the molar loading for a certain reaction.

## 3. Reactions Catalyzed by Pd/C

### 3.1. Reductions

It has already been established that a wide range of reducible functional groups, including carbon-carbon multiple bonds, carbon-oxygen double bonds, nitriles, and so forth, can be hydrogenated and reduced in the presence of a heterogeneous Pd/C catalyst. Currently, new Pd/C reduction methods are attracting the attention of chemists, and several relevant publications have already appeared [3]. The Pd/C-catalyzed reductions have also been shown to be effective in certain steps of total organic synthesis. In this section, we will discuss in detail the recent reduction reactions using Pd/C catalysis.

Patti and co-workers first investigated a Pd/C-catalyzed chemoselective hydrogenation of the nitrogen rings in quinoline derivatives to afford tetrahydroquinolines **2** [19]. Tetrahydroquinolines derivatives constitute a group of heterocycles that largely occur in natural products, whose main importance resides in their potential biological activities. The reaction proceeded well and the corresponding product was obtained at high yield (*r* = Ph, 88%) and high ratio (*r* = Ph, 95/5). As shown in Scheme 1, nitrogen cyclization was an important step. The hydrogenation process also involved the reduction of the nitryl group.

In 2013, Sugimura's group developed a synthetic protocol for the preparation of pioglitazone **5** from 5-alkylidene-2,4-thiazolidione **4** using heterogeneous Pd/C catalyst (Scheme 2) [20], which is a PPARγ agonist used in the treatment of individuals with type 2 diabetes. It was found that a lower amount of catalyst (5% Pd/C, 0.02 eq) than previously reported was effective [21]. Optimized conditions were successfully established and studies on kinetics were also reported in this literature.

**Scheme 1.** Pd/C-catalyzed synthesis of tetrahydroquinolines.

**Scheme 2.** Pd/C-catalyzed synthesis of pioglitazone.

Selective hydrogenation of multiple double-bonds is a major challenge in synthetic studies because there are several ways to reduce double bonds. Many factors such as the type of metal, the size of metal particles, the morphology, the addition of other metals and ligands, etc., affect this selectivity.

Su and colleagues reported the selective hydrogenation of citral **6** [22], which contains an isolated double bond and conjugated C=C, C=O bonds. All potential hydrogenation products are shown in Scheme 3. 3,7-Dimethyloctanal **9** was selected as the main target product in this work. The reaction was carried out in the presence of 1 wt% Pd/C in isopropanol at 303 K under hydrogen atmosphere and gave an excellent citral conversion of 100% and a high yield, of 96 %, of the desired product **12**.

The asymmetrical hydrogenation of prochiral compounds using modified Pd/C catalysts remains a challenging topic [23–26]. Inspired by the work of Izumi [27], the enantioselectivity of cinchona-modified palladium catalysts has been gradually discovered and explored. Nitta et al. made an important contribution to the enantioselective hydrogenation of α-phenylcinnamic acid (**15**, **16**) (PCA) on cinchonidine-modified palladium catalysts, and several publications on the asymmetric catalysis of this α,β-unsaturated carboxylic acid (Scheme 4), which is also a kind of chiral compound with wide applications, were published [28–31].

Sajiki and co-workers discovered chemoselective hydrogenation of olefins in the presence of diphenyl sulphide (Ph₂S) [32–34]. The high catalytic activity of Pd/C resulted in poor selectivity of hydrogenation. Moreover, catalyst poison was introduced into the reaction. Diphenyl sulphide (Ph₂S) was selected as a new type of catalyst poison to support Pd/C. Olefins were treated with hydrogen in MeOH at room temperature and gave chemoselective products in good to excellent yields (up to 100%) (Tables 1 and 2).

**Scheme 3.** Pd/C-catalyzed synthesis of 3,7-dimethyloctanal.

**Scheme 4.** Synthesis of α-phenylcinnamic acid via Pd/C catalysis.

**Table 1.** Application to Aromatic Halogen.

$$\text{substrate} \xrightarrow[\text{MeOH, H}_2(\text{1 atm}), \text{rt, 24h}]{\text{10\% Pd/C, Ph}_2\text{S(0.01 eq.)}} \text{product}$$

| Substrate | Product | Yield |
|---|---|---|
| **17** | no reaction | - |
| **18** | **19** | 98 |
| **20** | **21** | 97 |
| **22** | **23** | 90 [a] |

**Table 1.** *Cont.*

| | | |
|---|---|---|
| **24** | **25** | 99 |
| **26** | no reaction [b] | - |
| **27** | no reaction [b] | - |

[a] The yield was determined based on [1]H-NMR analysis and 10% of the starting material remained. [b] Diphenylsulfide (0.5 eq.) was added to the reaction mixture.

**Table 2.** Application to Benzyl Ester and *N*-Cbz Protective Group.

| Substrate | Product | Yield |
|---|---|---|
| **28** | **29** | 100 [a] |
| **30** | **31** | 100 [a] |
| **32** | **33** | quant |
| **34** | **35** | 95 |
| **36** | **37** | 94 |
| **38** | **39** | quant |
| **40** | **41** | 98 |
| **42** | **43** | 99 |

[a] The yield was determined based on [1]H-NMR analysis. It contains the error limit of ±5%.

Wang and Hu demonstrated a Pd/C-catalyzed hydrogenation to produce 4-benzylpiperidine hydrochlorides and α-alkyl-4-piperidinemethanol hydrochlorides **48** (Scheme 5) [35], which is an important intermediate in drug discovery. The reaction processes involved the reduction of the ketone, the hydrogenolysis of the C-O bond and the hydrogenation of the pyridine ring. It was assumed that the nitrogen atom in the pyridine ring coordinates with the Pd/C catalyst and deactivates the hydrogenation of pyridine ring, while the conversion to the corresponding pyridine salts solves this problem. ClCH2CHCl2 was finally selected as an additive to promote the reaction. Further studies found 4-pyridinemethanol hydrochloride derivatives **45** to be the key intermediate instead of the corresponding 4-alkylpiperidinol **47**; it was also found that **47** will never be converted into **48** because it is not a benzyl alcohol, and that its hydroxyl group could not be hydrogenolyzed under mild conditions.

**Scheme 5.** Synthesis of 4-benzylpiperidine hydrochlorides and α-alkyl-4-piperidinemethanol hydrochlorides via Pd/C catalysis.

Zhou and Wang developed a two-step Pd/C-catalyzed one-pot hydrogenation of carbonyl groups to produce methyl or methylene derivatives **52** (Scheme 6) [36]. *N*-tosylhydrazone **51** was an important intermediate in this process, which was replaced by *N*-tosylhydrazine **50** in the presence of MeOH at 60 °C. Optimization of the reduction of *N*-tosylhydrazone **51** was explored and the reaction proceeded with moderate to high yield in the presence of $K_2CO_3$.

**Scheme 6.** Reduction of *N*-tosylhydrazone by Pd/C catalysis.

Pihko's group reported a protocol for the preparation of enolsilane **54** using a Pd/C catalyst (Scheme 7) [37]. Treatment of α,β-unsaturated aldehydes or ketones **53** with $Et_3SiH$ in THF gave the corresponding enolsilane **54** at high yield and stereoselectivity. For α-substituted enals, the major products were Z-enolsilanes. β-substituted enals were converted to the corresponding E-enols. Several α,β-unsaturated cyclic ketones were employed and still gave the desired product in good yields. Further studies included the chemoselective hydrogenation of α,β-unsaturated carbonyl compounds **55** to produce aldehydes or ketones **56** over a similar Pd/C-$Et_3SiH$ catalyst system using inorganic acids.

**Scheme 7.** Reduction of α,β-unsaturated aldehydes or ketones by Pd/C catalysis.

Kim and co-workers developed a tributylamine-assisted hydrogenation of ketones **57** to produce the corresponding alkanes **58** using a Pd/C catalyst (Scheme 8) [38], another alternative approach to deoxyenative hydrogenation. Treatment of ketones **57** with Bu$_3$N and Pd/C in the presence of 4 Å MS in dioxane at 180 °C under argon atmosphere afforded alkane derivatives **58** in moderate to high yields (up to 90%).

**Scheme 8.** Reduction of ketones via Pd/C catalysis.

Macor's group found a Pd/C-catalyzed alkylation of α-methyltryptamines **61** that produced *N*-alkylated products at high yield (Scheme 9) [39]. Tryptamines **59** were treated with ketones or aldehydes to form the corresponding imines and then hydrogenated in the presence of hydrogen in EtOH to give *N*-alkylated amines. The application of this protocol was extended to other primary and secondary amines.

**Scheme 9.** Synthesis of N-alkylated tryptamines via Pd/C catalysis.

Pd/C has also been recognized as a clean and economical catalyst for the reduction of arylsulfonyl compounds to produce thiophenol derivatives **63** by Chen and co-workers

(Scheme 10) [40]. 2-Propanol was added as a hydrogen sources and solvent. The introduction of PPh$_3$ into the reaction system was found to be necessary as a ligand. Iodine was found to promote the reduction of arylsulfonic acid **62** by forming iodotriphenylphosphonium iodide, which improved the subsequent substitution process.

**Scheme 10.** Synthesis of thiophenol derivatives via Pd/C catalysis.

Beller et al. discovered a new hydrogenation strategy of aryl nitriles for the synthesis of primary amines **65** with Pd/C (Scheme 11) [41]. In this new approach, no toxic or hazardous reagents were used in the reaction system. The reaction proceeded well in the presence of HCOOH-NEt$_3$ (molar ratio 18.5:1) as hydrogen source/base in THF at room temperature or 40 °C. It was also evident that related functional groups such as amides and esters were not affected under the optimized conditions and the strategy was widely applicable to various aryl nitrile substrates **64**.

**Scheme 11.** Reduction of aryl nitriles via Pd/C catalysis.

*3.2. Coupling Reactions*

3.2.1. Suzuki–Miyaura Coupling Reaction

The Pd/C catalyst was first used by Buchecker and co-workers for the catalysis of Suzuki–Miyaura coupling reactions [42]. Treatment of aryl bromides or iodides with boronic acid in the presence of Pd/C and sodium carbonate at 80 °C gave the desired coupling products at high yield. It was not until a decade later that Sun and co-workers reported the coupling between aryl chlorides and aryl boronic acids [43]. The amount of literature on Pd/C-catalyzed Suzuki–Miyaura coupling reactions has increased significantly. Detailed studies on the mechanism and modifications of Pd/C-catalyzed Suzuki couplings were collected and reported by Felpin [44,45].

Sajiki's group developed a ligand-free Pd/C-catalyzed Suzuki–Miyaura coupling reaction under mild conditions (Scheme 12) [46]. This environmentally friendly reaction was carried out at room temperature in aqueous media under argon. The tested vinyl boronic acids **70** gave the desired coupling product, while few reports were published on the Suzuki–Miyaura coupling of substrates other than aryl boronic acids [47]. The reaction proceeded smoothly, even, in certain cases, under aerobic conditions.

R= 4-NO$_2$, CHO, CO$_2$Et, OH, OMe
2-Me, 2,6-Me$_2$
1-nap
R'=H, 2-OMe,3-OMe,4-OMe,4-COMe

26 examples
45–100%

R= 4-NO$_2$, OMe
2-Me
R"=H, Me,Cl,Ph

6 examples
80–91%

**Scheme 12.** Pd/C-catalyzed Suzuki–Miyaura coupling reactions of various aryl and vinyl boronic acids.

The other Suzuki coupling reaction of aryl chloride and boronic acid via Pd/C catalyst assisted with appropriate ligands was developed by Simeone and co-workers [16]. 2-Dicyclohexylphosphino-2′-methylbiphenyl phosphines (MePhos) and 2-dicyclohexylp-hosphino- 2′,4′,6′-trisopropylbiphenyl phosphines (XPhos) were selected as efficient ligands in the reaction system. The Pd/C-ligand catalytic system is not environmentally friendly, but the results of catalyst recycling experiments indicated that the Pd/C could be reused multiple times; this catalytic system also gave a much higher yield compared to the ligand-free catalytic system (Table 3).

**Table 3.** Ligand-free versus XPhos-assisted Pd/C-catalyzed Suzuki–Miyuara cross-couplings of aryl chlorides [a].

| R$^1$ | R$^2$ | Ligand Assisted Yield(%) [b] | Ligand Free Yield(%) [b] |
|---|---|---|---|
| 4-CH$_3$ | 2-CH$_3$ | 91 | 27 |
| 2-CH$_3$ | H | 71 | 15 |
| 2-CH$_3$ | 2-CH$_3$ | 80 | 7 |
| 2- OCH$_3$ | 2-CH$_3$ | 30 | 0 |
| 2,6-dimethyl | 2-CH$_3$ | 71 | 0 |
| 2-CH$_3$ | 4-CHO | 86 | n.d. |
| 2,6-dimethyl | 2-F | 70 | n.d. |
| 4-CH$_3$ | 2-F | 75 | n.d. |

[a] All reactions were performed with 0.8 mmol 4-chlroanisole **72**, 0.96 mmol phenylboronic acid **73**, 1.6 mmol XPhos, 0.2 mmol 5% Pd/C, in 5ml of 20:1($v/v$) DMA: H$_2$O at 80 for 24 h. [b] Isolated yield. n.d. means no data.

In 2010, Bhanage's group reported a ligand-free Pd/C-catalyzed Suzuki–Miyaura coupling strategy that provided biaryl and heteroaryl carbonyl compounds (**77, 79**) (Scheme 13) [48]. In this coupling reaction, three-components, such as heteroaryl/non-heteroaryl iodide (**75, 78**), carbon monoxide and boronic acids **76**, were involved. Notably, the treatment of these substrates in the presence of Pd/C at 100 °C afforded the corresponding coupling products in fair to excellent yield.

**Scheme 13.** Pd/C-catalyzed synthesis of biaryl and heteroaryl carbonyl compounds.

Sajiki and co-workers reported the hetero-Suzuki–Miyaura coupling reactions catalyzed by ligand-free Pd/C to afford both heterocyclic-alicyclic (**82**, **85**) and heterocyclic-heterocyclic biaryl derivatives **88** (Scheme 14) [7]. The reaction between heteroaryl/non-heteroaryl halide and heteroaryl boronic acids were all accommodated in the reaction and gave moderate to excellent yields in the presence of trisodium phosphate as base. The aryl halides involved in this work are mostly aryl bromides.

**Scheme 14.** Hetero-Suzuki–Miyaura couplings catalyzed ligand-free via Pd/C reduction.

In their follow up studies, Sajiki's group further disclosed solvent-free Suzuki coupling reactions via Pd/C catalyst [49]. The coupling reactions were taken place between solid aryl halides **89** and solid arylboronic acids **90**, in the presence of solid 10% Pd/C and $Cs_2CO_3$ or $K_2CO_3$, under absolute solid-phase conditions, and afforded moderate to excellent yield for most substrates (Scheme 15). Moreover, solvent free synthesis in the solid state is a valid approach to obtain biologically active heterocycle rings in ecofriendly way [50]. In addition, this group studied a solvent-free Pd/C-catalyzed heteroaryl coupling reaction between 2-bromopyridine **92** and *p*-methoxyphenylboronic acid **93** (Scheme 16).

**Scheme 15.** Solvent-free Pd/C-catalyzed synthesis of biphenyls.

R= 4-NO$_2$, CHO, OH, OMe, Me, OH, CN, NH$_2$;
   2-Me;
   1-Br-naphthalene, 4-Cl-nitrobenzene

R'=2-OMe,3-OMe,4-OMe,4-COMe, 4-Ac

18 examples
10–100%

**Scheme 16.** Solvent-free Pd/C-catalyzed synthesis of 2-arylpyridine.

A ligand-free Pd/C-catalyzed Suzuki–Miyaura reaction protocol for the coupling cyclic 2-iodo enones **95** with boronic acids **96** was developed by Felpin's group in 2005 [51]. This method was soon used for the synthesis of isoflavones. Tetrahydropyran (THP)-protected idochromanone was treated with arylboronic acid in aqueous dimethyl ether at 45 °C, giving the corresponding product at 73–94% yield. Further studies included the total synthesis of several isoflavones, such as 7-O-geranylformononetin **98**, Griffonianon D **99** and Conrauinon D **100**, via this coupling reaction (Scheme 17) [52].

6 examples
73–94%

7-O-geranylformononetin **98**
two steps from **95**, 69% overall

griffonianone D **99**
< 5% from **98**

conrauinone D **100**
two steps from **97**, 71% overall

**Scheme 17.** Synthesis of various isoflavones.

In the meantime, Felpin et al. developed a Pd/C-catalyzed Suzuki coupling of arene-diazonium tetrafluoroborate salts **101** with boronic acids **102** to synthesize biphenyls **103**

and terphenyl derivatives **104** (Scheme 18) [53]. The reaction underwent smoothly within several minutes even in the absence of any bases and ligands. Coupling products of halide-substituted arenediazonium salts could also be applicable to the subsequent Suzuki coupling reaction to construct terphenyls **104**. In addition, a one-pot protocol to synthesize terphenyls, which started with arenediazonium salts, was also developed.

R= 4-Br, 4-NO₂, 4-OMe, 3-CF₃, He

R'=H,3-NO₂,4-OMe,4-Cl, 3,4-dimethoxy

**Scheme 18.** Synthesis of biphenyl and terphenyl derivatives via Pd/C catalysis.

Liu's group succeeded in realizing an aerobic Pd/C-catalyzed ligand-free Suzuki coupling reaction under mild conditions (Scheme 19) [54]. The authors identified that the reaction was much quicker under air or oxygen than under nitrogen atmosphere. The protocol was applied to synthesize fluorinated liquid-crystalline compounds **110** under mild reaction conditions.

R= 4-CN, 4-CHO, 4-OAc, 4-CF₃,4-F,4-OH,4-OMe,4-CH₃,2-CN,S-OMe

R'=H,3-CH₃,4-CH₃,4-CN, 4-F

R=C₃H₇,C₅H₁₁

**Scheme 19.** Pd/C-catalyzed ligand-free Suzuki reaction.

Bora's group studied the Suzuki–Miyaura coupling of benzoyl chlorides **111** with arylboronic acids **112** for the synthesis of biaryl ketones **113** over heterogeneous Pd/C catalysts (Scheme 20) [55]. The reaction proceeded smoothly in the presence of sodium carbonate at 60 °C. The development of the carbon monoxide-free protocol for the synthesis of biaryl ketone derivatives contributed to the introduction of benzoyl chlorides as substrates.

**Scheme 20.** Synthesis of biaryl ketones from benzoyl chlorides via Pd/C catalysis.

### 3.2.2. Sonogashira Coupling Reaction

Guzman and co-workers presented the first Pd/C-catalyzed Sonogashira coupling reaction in 1990 [56]. By treating aryl bromide with phenylaceylene in the presence of Pd/C, phosphine, copper(I) iodide and triethylamine, the coupling product was obtained at high yield (53–88%). The tetrakistriphenylphosphinepalladium (0) [Pd(PPh$_3$)$_4$], a homogeneous palladium catalyst, was detected in the reaction mixture.

Pal's group found Terminal alkynes **115** were treated with *o*-halobenzenesulfonamide **114** to afford the target products for iodocyclization (Scheme 21) [57]. The coupling process proceeded smoothly when acetonitrile was chosen as the solvent in the presence of Et$_3$N, 10% Pd/C, PPh$_3$, and CuI (a ratio of 1:4:2) at 80 °C. Further studies showed that the Et$_3$N-Pd/C-PPh$_3$-CuI catalytic system was efficient in Sonogashira reactions.

**Scheme 21.** Pd/C-catalyzed Sonogashira coupling reaction.

Pal and co-workers applied the Sonogashira coupling protocol to the synthesis of isoquinolones **119** (Scheme 22) [58]. The coupling reaction between 2-iodobenzoylazides **117** and terminal alkynes **118**, followed by an intramolecular Schmidt reaction, afforded corresponding target products in moderate to excellent yields. The use of EtOH was key for the stereoselective formation of isoquinolones **119** instead of five-membered isoindolones. The Pd/C-CuI-PPh$_3$ catalytic system was still significant for this new methodology in both the Sonogashira coupling and Schmidt cyclizing processes. A possible mechanism involves the in situ generation of o-alkynyl azido benzene **120** via Pd/C-mediated Sonogashira coupling, wherein Pd (II) generated **121** from Pd (0) activates the C-C triple bond, which undergoes nucleophilic addition by the proximal nitrogen of the azide. The subsequent loss of dinitrogen produces the cationic intermediate **123**, which, as a result of the hydride shift followed by the regeneration of Pd (0), finally affords the isoquinolone **119**.

Liu and co-workers disclosed a phosphine-free Pd/C-catalyzed carbonylative Sono-gashira coupling reaction (Scheme 23) [59]. This research was the first report on car-bonylative Sonogashira coupling reactions achieved by heterogeneous catalysts. The three-component reactions were found to go smoothly in the presence of triethylamine in toluene at 130 °C. Only aryl iodides **125** showed reactivity in this investigation.

Scheme 22. Synthesis of isoquinolin-1(2H)-one via Pd/C catalysis.

Scheme 23. Phosphine-free Pd/C-catalyzed Sonogashira coupling reaction.

Sajiki's group reported a ligand-, copper-, and amine-free Pd/C-catalyzed Sonogashira coupling reaction (Scheme 24) [60], and this was found to be applied to broader aryl iodide substrates as compared to Zhang's work, which was mentioned previously [9]. The reaction was conducted by only a small (0.4 mol%) amount of 10% Pd/C catalysts in the presence of Na$_3$PO$_4$ at 80 °C. In this study, 50% *i*-PrOH was found to dissolve Na$_3$PO$_4$ well, and Na$_3$PO$_4$ was believed its pKa value is nearly the same as amines universally used for the traditional Sonogashira coupling reaction, such as triethylamine or diethylamine. The protocol could also be applicable in an aerobic atmosphere in certain cases.

Pal's group disclosed another Sonogashira coupling process to prepare 6-oxopyrroloquinolines **133** from quinoline derivatives **132** and terminal alkynes **131** (Scheme 25) [61,62]. The Pd/C, PPh$_3$, and CuI (a ratio of 1:4:10) catalyst system was again proved efficiently in this Sonogashira reaction in refluxing ethanol. The reaction proceeded well with various substrates. An easy and inexpensive strategy was developed to access target products.

**Scheme 24.** Synthesis of phenylalkyne derivatives via Pd/C reduction.

**Scheme 25.** Synthesis of 6-oxopyrroloquinolines via Pd/C reduction.

Bakherad and co-workers studied the synthesis of 2-benzylimidazo[1,2-α]pyrimidines **140** in which Pd/C-catalyzed Sonogashira coupling reaction played an important role (Scheme 26) [63]. Substrates were prepared by treating 2-aminopyrimidine **134** with propargyl bromide **135** in refluxing acetonitrile. The best results were obtained when the reactions were carried out with 5 mol % of Pd/C and 20 mol % of PPh$_3$ at 95 °C in the presence of K$_2$CO$_3$ as a base. The new protocol does not employ hazardous copper iodide and still provided corresponding products in moderate to high yields.

**Scheme 26.** Synthesis of 2-2benzylimidazo[1,2-α]pyrimidines via Pd/C reduction.

### 3.2.3. Hiyama Coupling Reaction

In 1988, Hiyama and Hatanaka discovered the coupling of organosilanes with organic halides in the presence of tris(diethy1amino)sulfonium difluorotrimethylsilicate (TASF) and allylpalladium chloride dimer to synthetically form C-C bonds with chemo- and regio-selectivity [64]. The heterogeneous Pd/C catalyst was first reported to be introduced into the Hiyama coupling reaction by Novak et al. in 2010 (Scheme 27) [65]. The coupling protocol required the assistance of 4 mol% PPh$_3$ as a ligand and tetrabutylammonium

fluoride (TBAF) as a fluoride source in the catalytic system. Moderate to good yields (45–76%) of desired products were obtained after condition optimization and substrate extension work.

R= 4-OMe, 4-CHO, 4-CH$_3$, 4-CF$_3$,4-OCF$_3$
3-NH$_2$

8 examples
45–76%

**Scheme 27.** Synthesis of biphenyl derivatives via Pd/C reduction.

Sajiki and co-workers reported the Pd/C-catalyzed Hiyama coupling reaction of various aryl iodides or aryl bromides **144** with substituted aryltriethoxysilanes **145** (Scheme 28) [66–68]. The protocol required only a small amount of 10% Pd/C (0.5 mol%) assisted by 1 mol% tris(4-fluorophenyl)phosphine as ligand in the presence of TBAF in toluene at 120 °C. The addition of small amount of water in the reaction system was found to enhance the reaction efficiency. The hydrolysis of aryltriethoxysilane was promoted by water to give corresponding arylsilanol mixtures, leading to a decrease in electrophilicity in silicon atom centers, which promoted the subsequent formation of a silicate complex.

X= I or Br

R= 4-NO$_2$, 4-OMe, 4-Ac, 3-Ac,2-OMe

R'=H,4-Cl, 4-Me

21 examples
31–90%

**Scheme 28.** Synthesis of biphenyl derivatives via Pd/C reduction.

3.2.4. Other C-C Coupling Reaction

The Heck coupling reaction, known as the first heterogeneous Pd/C coupling reaction, has been fully developed since its discovery in 1972 [2]. In Heck coupling, aryl halides were used with vinyl derivatives to obtain coupling products, but they were not as efficient as those carried out with homogenous catalysts. The protocol was further developed and the mechanism was explored. Hallberg's group optimized the reaction conditions and studied the regio- and stereoselectivity of the coupling process [69]. Arai and co-workers also focused on optimization work and contributed much to the elucidation of the mechanism of the Heck coupling procedure [70–73]. Köhler and coworkers first developed styrene by means of a Heck coupling reaction [74–76]. Several recent studies on the Heck coupling reaction are listed as follows.

The Heck coupling strategy was applied to the aminocarbonylation of aryl iodides by Bhanage's group in 2008 (Scheme 29) [77]. In the coupling process, *N,N*-dimethylformamide **148** (DMF) was treated with aryl iodides to avoid the use of carbon monoxide as a source of carbonyl groups. POCl$_3$ was included in the reaction system to assist in the preparation of imminium salt from DMF, which underwent nucleophilic addition with aryl palladium iodide.

**Scheme 29.** Synthesis of *N,N*-dimethylbenzamide derivatives via Pd/C reduction.

Djakovitch's group investigated the synthesis of diethyl 2-(aryl)vinylphosphonate via a Pd/C-catalyzed direct Heck coupling reaction (Scheme 30) [78]. Diethyl vinyl phosphonate **151** was coupled with various aryl halides (Br or I) or heteroaryl halides **150** in DMF at 100–140 °C. The reactants were all commercially available and high conversions were observed. In the cases of aryl iodides or safely activated aryl bromides, only 0.25 mol% palladium was required in the reaction system.

**Scheme 30.** Synthesis of styrylphosphonate derivatives via Pd/C reduction.

Sajiki's group studied the Pd/C-catalyzed homocoupling of terminal alkynes **153** to produce 1,2-diynes **154**, which is known as the Glaser coupling reaction (Scheme 31) [79]. Previous researchers studied the effect of copper catalysts and homogeneous palladium-copper catalysts on the homocoupling process [62]. Sajiki and co-workers worked on heterogeneous Pd/C-catalyzed protocol. Subsequent optimization work showed that only 0.01 mol% of 1%Pd/C was required with the support of 3 mol% CuI in DMSO as a solvent. Using this strategy, a variety of diyne derivatives **154** could be prepared in good to excellent yields.

**Scheme 31.** Synthesis of diyne derivatives via Pd/C catalysis.

Qi's group disclosed a Pd/C-catalyzed ethanol-promoted Ullmann-type homocoupling of aryl halides to provide biaryls (Scheme 32) [80]. An increasing aryl halides and biaryls conversion was observed with the increasing of the amount of ethanol, while the excess ethanol-induced self-reduction product suppressed the production of biaryls. Cesium fluoride was selected as efficient base in the optimization process and the reaction went smoothly in DMSO at 120 °C. The amount of ethanol varied from 0.86eq. to 8.56eq. as different aryl halides were selected as reactants.

**Scheme 32.** Synthesis of biphenyl derivatives via Pd/C reduction.

Leblond and co-workers discovered the couplings that involved the use of potassium *p*-tolyltrifluoroborate **157** instead of boronic acid and boronic esters (Scheme 33) [81]. The protocol could only be applied to aryl bromides and iodides, while a green, easily prepared and stable salt was introduced as a substrate. The reaction proceeded well under mild conditions (ethanol/water = 5:1, Pd/C 10 mol% and $K_2CO_3$ at 50 °C) and the corresponding products were obtained in moderate to good yield (42–85%).

**Scheme 33.** Synthesis of biphenyl derivatives via Pd/C reduction.

Zhu's group developed a cyclization coupling to obtain 2,3′-bisindoles **162** (Scheme 34) [82]. The strategy involved cross-coupling of two *o*-alkynylanilines (**160**, **161**) and was conducted in the presence of *n*Bu₄NBr and acetic acid. The possible mechanism for the formation of bisindole is shown as follows. Selective aminopalladation of **160** would afford the *s*-indolylpalladium (II) intermediate **163** that, upon ligand exchange, would provide **164**. Reductive elimination, followed by *N*-demethylation, would then provide 3-alkynylindole **165** and Pd⁰. Oxidation of Pd⁰ to Pd^II followed by a second aminopalladation would afford **166** that, upon demetallation, was expected to produce the bisindole **162**.

Glorius and co-workers developed an efficient protocol for the arylation of benzo[b]thiophene **167** catalyzed by heterogeneous Pd/C catalyst (Scheme 35) [83]. Treatment of benzo[b]thiophene **167** with chlorobenzene **168** in the presence of Pd/C catalysts at 150 °C was found to provide corresponding 3-arylbenzo[b]thiophene **169** with high C3/C2 selectivity. The yield reached 60% after standard optimization of the reaction conditions. To further improve the yield of the reaction, CuCl was found to enhance reactivity by activating the Lewis acid of benzo[b]thiophene **167**. Further study suggested a high selectivity in an intermolecular competition with pyridine **171** and butylthiophene **170** when treated with chlorobenzene under standard optimized reaction conditions (Scheme 36).

**Scheme 34.** Pd/C-catalyzed synthesis of indole derivatives.

**Scheme 35.** Synthesis of benzo[b]thiophene derivatives via Pd/C reduction.

**Scheme 36.** Intermolecular competition experiment.

In 2005, Seki et al. reported a Pd/C-catalyzed cyanation of aryl halides via zinc reagents [84]. Based on the cyanation work of Seki's group, Yu et al. explored a Pd/C cyanation method to produce aryl cyanide **176** from aryl halides **175** and extended the substrates (Scheme 37). The fact that cyanide ion could easily interact with palladium caused the deactivation of the Pd/C catalyst. Weakly ionizable M-CN bonds, such as zinc cyanide, were selected as cyanide ion sources assisted by Zn $(CHO)_2 \cdot 2H_2O$. 1,1'-bis(diphenylphosphino)ferrocene (dppf) was selected as an efficient ligand. The reaction was processed in dimethylacetamide at 100–120 °C. The protocol could be applied to various aryl bromides and active aryl chlorides. The Pd/C-catalyzed cyanation of het-

eroaryl halides was also explored and corresponding cyanation products were obtained in moderate to good yields.

**Scheme 37.** Synthesis of benzonitrile derivatives via Pd/C reduction.

3.2.5. Other C-X Coupling Reaction

Our group discovered an environmentally friendly Pd/C-catalyzed coupling for formation of C-S bond to produce aryl sulfides **179** without any hazardous ligands (Scheme 38) [85]. Various homogeneous Pd catalytic systems with different ligands as well as other metal-based catalysts were reported to successfully realize the coupling reaction since Migita et al. first found Pd-catalyzed C-S coupling between aryl iodides and thiols [86]. Optimization work indicated that coupling reaction went smoothly in the presence of KOH in DMSO under nitrogen at 110 °C. Further investigation also found that diaryl sulfides were obtained in moderate yield using aryl bromides or aryl chlorides bearing the electron-withdrawing groups. Recycled Pd/C did not show a marked loss of catalytic activity within the fifth reuse of the filtered catalyst.

**Scheme 38.** Synthesis of aryl sulfides via Pd/C reduction.

A clean and inexpensive Pd/C-catalyzed allylic substitution protocol to form various C-N, C-O or C-S bonds in an aqueous system was explored by Felpin's group [87]. Allylic acetates (**180, 182, 184**) were selected as substrates and were treated with heteroatom-contained nucleophiles in the presence of $PPh_3$ and Pd/C in water, and corresponding substituted products (**181, 183, 185**) were obtained in moderate to high yields (Scheme 39). The base was also required in certain cases. A comparison of two catalysts between homogeneous Pd $(PPh_3)_4$ and heterogeneous Pd/C was studied and Pd/C showed great advantages; in addition to higher efficiency being observed following catalysis by Pd/C, an excellent chemoselective preference to obtain our desired product was also detected.

**Scheme 39.** Allylic substitution via Pd/C reduction.

Li's group developed a protocol for the formation of intramolecular C-S bond to provide 2-substituted benzothiazoles **187** from *o*-iodothiobenzanilide **186** (Scheme 40) [88]. The reaction was conducted in the presence of 2 mol% Pd/C at room temperature in DMF and did not require any ligands and other additives. The reaction condition could be applied to a wide range of substrates with different substitutions on the phenyl ring. The model reaction was further extended to the synthesis of 2-alkyl and 2-aminobenzothiazoles (**189**, **191**) besides 2-arylbenzothiazoles.

R$^1$= H,4-Cl, 4-OMe, 4-Me,2-I

R$^2$=H,4-OMe, 4-Cl,4-CN

**Scheme 40.** Synthesis of 2-substituted benzothiazoles via Pd/C reduction.

Rummelt and Ranocchiari disclosed the formation of P-C bonds via the Pd/C-catalyzed coupling process to produce tertiary phosphine oxides (Scheme 41) [89]. Treatment of 4-iodobenzoic acid **192** with diphenylphosphine oxide **193** in the presence of K$_2$CO$_3$ in water afforded the desired product **194** in a high yield. The green and economic protocol did not involve organic solvent, ligands, and other hazardous metals in the reaction system. Further investigation indicated that the use of microwave heating, instead of conventional heating, promoted the corresponding reactions. For some simple halogenated benzoic acids, such as 3- and 4-bromobenzoic acid and 3- and 4-iodobenzoic acid, iodo derivatives

were found to yield fewer products due to their easier tendency to dehalogenation under reaction conditions. For 2-halogen benzoic acids, only traces of corresponding products were obtained due to steric hindrance.

X= I or Br or Cl

R= H,3-NH$_2$, 4-OMe, 4-Me

**Scheme 41.** Synthesis of tertiary phosphine oxides via Pd/C reduction.

Among all the carbon-heteroatom coupling processes, C-N bond formation attracted much attention and several kinds of studies on this issue have been published since Buchwald and Hartwig first independently developed a palladium-catalyzed aromatic amination protocol in 1995 [90,91]. It was Djakovitch's group that first disclosed the application of heterogeneous Pd/C catalysts in the preparation of arylamines(**197**, **198**) (Scheme 42) [92]. Treatment of bromoanisole **195** with piperidine **196** in the presence of *t*-BuONa produced both para- and meta-adducts. The involvement of benzyne intermediates during the reaction process led to this poor regioselectivity.

**197**:**198** = 1.7:1, 81%

**Scheme 42.** Aromatic amination via Pd/C reduction.

Novak's group polished the Buchwald–Hartwig amination process and employed aryl iodides **199** [93], which were not generally used before and were the least successful aromatic sources in this amination (Scheme 43). Optimization work showed that the reaction of aryl iodide **199** and amine **200** went smoothly when 1 mol% 10% Pd/C Selcat Q6 and 1 mol% (2-biphenyl)dicyclohexylphosphine (DiCyJohnPhos) were added as catalyst and ligand in the presence of *t*-BuONa in *t*-BuOH at 80 °C under argon. Recycled Pd/C catalyst was also tested and only a small degree of activity was lost.

X= I or Br

R= H,4-Ac, 4-OMe, 4-Me,2-nap

R'=alkyl, aryl group

**Scheme 43.** Synthesis of secondary amine derivative via Pd/C reduction.

In 2013, Sajiki's group disclosed the synthesis of indoles **204** via an intramolecular amination by applying the Buchwald–Hartwig amination strategy (Scheme 44) [94]. The

reaction employed 2-bromophenethylamines **202** treated with a 10% Pd/C- dppf- *t*-BuONa catalytic system. The neutralization by acetic acid after cyclization was found to promote the aromatization process. Pd/C-catalyzed coupling between indoles **205** and aryl halides **206** to provide *N*-arylindoles **207**, as well as a one-pot protocol to produce *N*-arylindoles directly from 2-bromophenethylamine, was also developed.

**Scheme 44.** Synthesis of indoles via Pd/C reduction.

## 4. Conclusions

In this review, we summarized the advances in Pd/C-catalyzed reactions, mainly reduction reactions and cross-coupling reactions. Due to its catalytic efficiency, easy recyclability, and diverse catalytic modes, the Pd/C catalyst has found, and will continue to find, valuable applications in the catalysis of organic reactions. It is worth pointing out that there is wide variability in the efficiency of commercial sources of Pd/C, resulting in significant differences in selectivity, reaction times, and yields [95]. On the other hand, it is still a significant challenge to explore the solution for the Pd/C catalyst's progressive deactivation upon successive reuses when, due to palladium leaching, the catalyst is hard recycled after a few times, and thus, it becomes inactive or much less active. Therefore, far more efforts are needed to develop sustainable chemistry and more efficient catalytic reagents, to understand their catalytic mechanism, and to discover new reactions that can be efficiently catalyzed by such heterogeneous catalysts.

**Author Contributions:** Conceptualization, Z.M., X.L.; formal analysis, H.G.; writing—original draft preparation, Z.M.; writing—review and editing, H.G.; visualization, Z.M., H.G., X.L.; supervision, X.L.; funding acquisition, X.L. All authors have read and agreed to the published version of the manuscript.

**Funding:** Financial support from Leading Talents of Special Support Program of Zhejiang Province High-level Talents (2020R52008) and Center of Chemistry for Frontier Technologies of Zhejiang University is gratefully acknowledged.

**Data Availability Statement:** Not applicable.

**Conflicts of Interest:** The authors declare no conflict of interest.

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
