# Peer review of "Recent Advances of Pd/C-Catalyzed Reactions"

_catalysts, doi:10.3390/catal11091078_

Round 1

Reviewer 1 Report

In this manuscript, the authors describes a general overview of Pd/C-Catalyzed Reactions, in particular reduction and coupling reactions. Pd/C is a common heterogeneous catalyst that can be applicated in countless reactions, in fact in literature are present more than five thousand articles that describes the reactions catalyzed by this key system, confirming the important role played in organic synthesis. Generally, the homogeneous palladium catalyst is very useful in organic chemistry but its recovery for further applications it's not always easy, so the use of this catalytic system represents an unsolved problem in industrial applications due to very expensive recovery operations and disposal. In this context, the heterogeneous Pd/C catalyst is a valid alternative because it has an higher catalytic efficiency and can be easily recovered by simple filtration from reaction environment. In other terms, its small loss catalytic activity and its high easily recovery permit the application for more catalytic cycles to give an industrial synthetic process remarkably little polluting. Moreover, Pd/C in association with molecular hydrogen is the best reducing system used in industrial organic processes because, in some cases, by acting on the reaction parameters, mainly temperature and pressure of the system, it is possible to regulate the degree of reduction of the compounds that react. In fact, even today, chemoselectivity in reduction reactions is an open and highly investigated field. In addition, Pd/C is an efficient catalytic system largely used for the formation of new C-C bonds in organic synthesis through coupling reactions. Based on these considerations, this review aims to collect and describe the most recent works concerning the reduction and coupling reactions catalyzed by Pd / C. Therefore, this review can represent a valid help to the scientific community that carries out scientific research on heterogeneous-catalyzed reduction and coupling reactions.

Review is well structured in introduction, heterogeneous catalyst description, reactions catalyzed with Pd/C and conclusion, but some modification are necessary before publication of manuscript.

In according with the authors, I have some general suggestions that must be adopted in the manuscript:

  1. Introduction part must be improved by explaining how the review was divided, what is the essential aspects highlighted in manuscript and the time frame of the publications considered.
  2. In the manuscript, every single molecule and every single class of molecules must be numbered with a bold number. This number must be entered both in the text and in the reaction schemes. The same molecule and the same class of molecules must have the same number in every part of the manuscript.
  3. In the manuscript, each reaction scheme must report the reaction conditions above and below the reaction arrow.
  4. In the manuscript, each reaction scheme should report the number of examples and the range of reaction yields. For example, if twenty molecules with yields ranging from 30% to 96% have been synthesized in an article, in the reaction scheme, the wording "20 examples" must be reported under the product and 30-96% must be reported under it.
  5. In the manuscript, you must indicate all reactions conducted on industrial scale.
  6. If possible, explain the catalytic efficiency in the reactions by indicating the number of recycles.
  7. Conclusions part must be improved.

Also, I have attached a file with specific suggestions that authors need to follow.

In conclusion, the manuscript have excellent novelty and for this reason I suggest the publication after minor revisions.

Author Response

Comment 1: Introduction part must be improved by explaining how the review was divided, what is the essential aspects highlighted in manuscript and the time frame of the publications considered.

Response: This review has highlighted the background and recent progress of Pd/C catalysis. Particularly, application has been emphasized to reactions involving reduction reactions and cross-coupling reactions. We have modified some part of the introduction part as highlighted with “in this review we have examined recent advances in the chemistry of Pd/C-catalyzed reactions such as reduction reactions and cross-coupling reactions, because these reactions are widely used for organic synthesis, they also provide a guiding principle for designing the route to organic chemists.”

Comment 2: In the manuscript, every single molecule and every single class of molecules must be numbered with a bold number. This number must be entered both in the text and in the reaction schemes. The same molecule and the same class of molecules must have the same number in every part of the manuscript.

Response: Done. All are checked and corrected.

Comment 3: In the manuscript, each reaction scheme must report the reaction conditions above and below the reaction arrow.

Response: Done. All are checked and corrected. Each reaction scheme reported the reaction conditions above and below the reaction arrow

Comment 4: In the manuscript, each reaction scheme should report the number of examples and the range of reaction yields. For example, if twenty molecules with yields ranging from 30% to 96% have been synthesized in an article, in the reaction scheme, the wording "20 examples" must be reported under the product and 30-96% must be reported under it.

Response: Done. All are checked and corrected. Each reaction scheme reported the number of examples and the range of reaction yields.

Comment 5: In the manuscript, you must indicate all reactions conducted on industrial scale.

Response: In this review, we mainly introduce the application of Pd/C-Catalyzed reactions in the laboratory, and the error is corrected.

Comment 6: If possible, explain the catalytic efficiency in the reactions by indicating the number of recycles.

Response: Done. The leached Pd nanoparticles may remain in solution and be separated from the products due to solubility differences, the catalyst is hard recycled after a few times.

Comment 7: Conclusions part must be improved.

Response: Done. We make the special supplement in Conclusions section with “It is worth pointing out that there is wide variability in the efficiency of commercial sources of Pd/C resulting in significant differences in selectivity, reaction times, and yields.” and “Therefore, much more efforts are needed to develop sustainable chemistry and more efficient catalytic reagents, to understand their catalytic mechanism, and to discover new reactions that can be efficiently catalyzed by such heterogeneous catalysts.”

Reviewer 2 Report

The review entitled “Recent Advances of Pd/C-Catalyzed Reactions” show the use of these catalysts for reduction reactions and cross-coupling reactions in organic synthesis. Even though the review shows interesting results, the discussion is really poor. The authors describe the results but there is neither further elaboration, poor comparison with current literature. The authors don't have to list everything in the field but discussion is very important for a review paper. There is a lot of information being summarized. Readers can easily get confused or lose track. Therefore, you need a lot of discussion to help readers compare the pros and cons of each reaction and then they will understand why people do this or that. In the same way it is important indicate the concentration of Pd used in each reaction since this is the paramount importance for the implementation of the catalysts. Finally, a short part of the review should show the characterization studies and some results in this field.

Additionally, for this review the authors should show the use of Pd/C catalysts in the inorganic synthesis since they are used in the production of H2 from different routes.

In short, even though the premise of this review is very interesting, the discussion of the parameters and results must really be improved to reach enough quality to be accepted.

Author Response

Comment : You need a lot of discussion to help readers compare the pros and cons of each reaction and then they will understand why people do this or that. In the same way it is important indicate the concentration of Pd used in each reaction since this is the paramount importance for the implementation of the catalysts.

Response: Done. All are checked and corrected. The concentration of Pd used in each reaction indicated.

Comment : a short part of the review should show the characterization studies and some results in this field.

Response: Done. We make the special supplement in Conclusions section with “It is worth pointing out that there is wide variability in the efficiency of commercial sources of Pd/C resulting in significant differences in selectivity, reaction times, and yields.” and “Therefore, much more efforts are needed to develop sustainable chemistry and more efficient catalytic reagents, to understand their catalytic mechanism, and to discover new reactions that can be efficiently catalyzed by such heterogeneous catalysts.”

Comment : for this review the authors should show the use of Pd/C catalysts in the inorganic synthesis since they are used in the production of H2 from different routes.

Response: This review has highlighted the background and recent progress of Pd/C catalysis. Particularly, application has been emphasized to reactions involving reduction reactions and cross-coupling reactions in organic synthesis. 

 Comment : the discussion of the parameters and results must really be improved to reach enough quality to be accepted.

Response: Each reaction scheme must report the reaction conditions above and below the reaction arrow. Each reaction scheme reported the number of examples and the range of reaction yields. we add some discussion of the parameters and results.

Reviewer 3 Report

The authors (Z. Mao, et al.) reported an overview of Pd/C catalyzed reactions including recent works.  This review manuscript covered not only reductions of C-C multiple bonds and other carbon-heteroatom multiple bonds, but a wide range of coupling reactions.  If possible, the authors should be described in a little more detail the physical and chemical features of Pd/C including Pd/C(en) and Pd/Fib.

The manuscript is well organized and can publish on Catalysts after the following small revisions.

  1. Some of the words were written in normal type-face which should be shown in italic face, e.g. N-Cbz…, Z-enolsilane…, N-tosyl… and so on (lines 129, 146, 157…). The authors should check and correct all of them in the manuscript.
  2. Some chemical structures are deformed in Table 2.
  3. Text captions and atom labels should be the same size in Scheme 6.
  4. Line 216 on page 9, 2-dicyclo… should be started by capital.
  5. Line 351 on page 16, et. Al… should be changed to et. a..
  6. Line 359 on page 16, aryltriethyxy should be changed to aryltrietho

Author Response

Comment 1: Some of the words were written in normal type-face which should be shown in italic face, e.g. N-Cbz…, Z-enolsilane…, N-tosyl… and so on (lines 129, 146, 157…). The authors should check and correct all of them in the manuscript.

Response: Done. All are checked and corrected

Comment 2. Some chemical structures are deformed in Table 2.

Response: Done. All are checked and corrected

Comment 3. Text captions and atom labels should be the same size in Scheme 6.

Response: Done. All are checked and corrected

Comment 4. Line 216 on page 9, 2-dicyclo… should be started by capital.

Response: Done. The error is corrected.

Comment 5. Line 351 on page 16, et. Al… should be changed to et. a..

Response: Done. The error is corrected.

Comment 6. Line 359 on page 16, aryltriethyxy should be changed to aryltrietho

Response: Done. The error is corrected.

Round 2

Reviewer 2 Report

This paper has been improved.  This is higher quality version in order to deserve publication.